# DATA-DRIVEN FEATURE SAMPLING FOR DEEP HYPERSPECTRAL CLASSIFICATION AND SEGMENTATION

## ABSTRACT

The high dimensionality of hyperspectral imaging forces unique challenges in scope, size and processing requirements. Motivated by the potential for an in-the-field cell sorting detector, we examine a *Synechocystis sp.* PCC 6803 dataset wherein cells are grown alternatively in nitrogen rich or deplete cultures. We use deep learning techniques to both successfully classify cells and generate a mask segmenting the cells/condition from the background. Further, we use the classification accuracy to guide a data-driven, iterative feature selection method, allowing the design neural networks requiring 90% fewer input features with little accuracy degradation.

## 1 INTRODUCTION

Hyperspectral confocal fluorescence microscopy and hyperspectral imaging are powerful tools for the biological sciences, allowing high-content views of multiple pigments and proteins in individual cells within larger populations. As the technology has advanced in speed and ease of use, it is has become practical to think of applications such as high-throughput screening, or understanding heterogeneous cell response to changing environmental conditions, where one might want to identify cells of certain characteristics including phenotype, pigment content, protein expression, as determined by their spatially resolved fluorescence emission for subsequent analysis. Although a few researchers have used classification techniques such as support vector machines (Rajpoot & Rajpoot, 2004) to identify cells of that exhibit similar spectral emission characteristics, the majority of the analysis of hyperspectral images has been exploratory—developing spectral models for identifying the underlying spectral components (Vermaas et al., 2008; Collins et al., 2012; Murton et al., 2017).

In this work, we employ deep artificial neural network algorithms to classify individual cyanobacterial cells based on their hyperspectral fluorescence emission signatures. Such deep learning methods have increasingly seen extensive use in conventional image processing tasks with relatively low numbers of channels (such as processing RGB images) (LeCun et al., 2015), however their utility in tasks with larger numbers of sensors, such as hyperspectral systems, remains an area of active research. In particular, in biological systems, non-trivial processes may yield complex interactions that can be detected through hyperspectral imaging that are in addition to the long-acknowledged challenges of automated data processing of spatial structure.

In addition to classifying the experimental effects on individual cells, we show how this method can help identify which spectral wavelengths are most useful for the classification. Importantly, the feature selection information could allow customized sensors to be designed for specific applications. This work demonstrates that this technique is suitable for real-time image analysis and high-throughput screening of heterogeneous populations of cyanobacterial cells for differentiating environmental response. The method can be further extended to other cell populations or complex tissue containing multiple cell types.

## 2 METHODS

### 2.1 DATASET

Cyanobacterial culture, hyperspectral confocal fluorescence microscopy, spectral image analysis, and single cell analysis have been described fully in a previous publication (Murton et al., 2017). In brief, *Synechocystis sp.* PCC 6803 cells were grown photoautotrophically in BG11 medium with 1.76 M NaNO$_3$ (nitrogen containing cultures) or where 1.76 M NaCl was substituted for the NaNO$_3$ (nitrogen deplete cultures). Cultures were maintained under cool white light ($30\mu$mol photon m-2 s-1, constant illumination) at 30°C with shaking (150 rpm). Samples were obtained at 0, 24, 48 hours for imaging studies. Fig. 1 in Murton et al. shows the experimental design. A small amount of concentration cyanobacterial cell solution ($8\mu$L) was placed on an agar-coated slide. After a brief settling time (1min) the slide was coverslipped and sealed with nail-polish. Imaging was performed immediately. Hyperspectral confocal fluoresce images were acquired using a custom hyperspectral microscope (Sinclair et al., 2006) with $3\mu$W of 488nm laser excitation and a 60x oil objective (NA 1.4). Spectra from each pixel were acquired with an electron multiplying CCD (EMCCD) with 240ms dwell times/pixel and the image was formed by raster scanning with a step size of $0.12\mu$m. Hyperspectral images were preprocessed as described in Jones, et al. (Jones et al., 2012) to correct for detector spikes (cosmic rays), subtract the detector dark current features, generate cell masks that indicate background pixels, and perform wavelength calibration. To discover the underlying pigments relevant to the biological response to nitrogen limitation, multivariate curve resolution (MCR) analysis (Haaland et al., 2015) was performed using custom software written in Matlab. Alternatively, the preprocessed hyperspectral images were subjected to classification (subject of this paper).

### 2.2 DATA FORMATTING

We performed image classification on the hyperspectral images both on individual pixels, which thus did not incorporate spatial information, and on whole images, which combined cell masking with classification. For training, we only used images from the 24hr timepoint.

For pixel classification, to form the training, validation, and testing datasets, we first divide each of the traces and corresponding masks (generated using an automated cell segmentation routine based on a modified marker watershed transform with input from a skilled user) into individual pixels, see Fig. 1(a). This provides roughly 44k 512-dimensional vectors per trace, each with a corresponding ground truth mask representing Background (BG), Cell Grown in Nitrogen Containing Culture (N$^+$) or Cell Grown in Nitrogen Deplete Culture (N$^-$) respectively. Since the vast majority of the pixels are background pixels, we perform uniform random undersampling to obtain a dataset that contains roughly 30k samples of each class. We randomly select $80\%$ for training, $10\%$ for validation, and $10\%$ for testing.

For joint cell masking and classification, we generated roughly $9,000$ $48 \times 48$ pixel chips using standard data augmentation techniques (crops, reflections, rotations) on both the original 24hr images and the ground truth masks. The images were undersampled during generation ensuring that the vast majority (roughly 90%) contained non-trivial masks. One-twentieth of the dataset was set aside for validation. Similar chips were generated from the 48hr dataset for testing.

### 2.3 DENSELY CONNECTED NETWORK FOR PIXEL CLASSIFICATION

Pixels from experimental images were classified into one of three categories: Background (BG), Cell Grown in Nitrogen Containing Culture (N$^+$) or Cell Grown in Nitrogen Deplete Culture (N$^-$). To perform pixel classification, we use a densely connected feed-forward neural network pictured in Fig. 2(a). A dropout layer helps prevent overfitting, and the network is trained using an Adam optimizer (Kingma & Ba, 2014). Rectified linear activation is used for the dense layers. Hyperparameter optimization was accomplished using hyperas (pum) which is a wrapper for hyperopt (Bergstra et al., 2013) on keras (Chollet et al., 2015). For a comparative baseline, we also performed the classification using 100-tree, 10-simultaneous-feature random forests, implemented in Scikit-learn (Pedregosa et al., 2011).

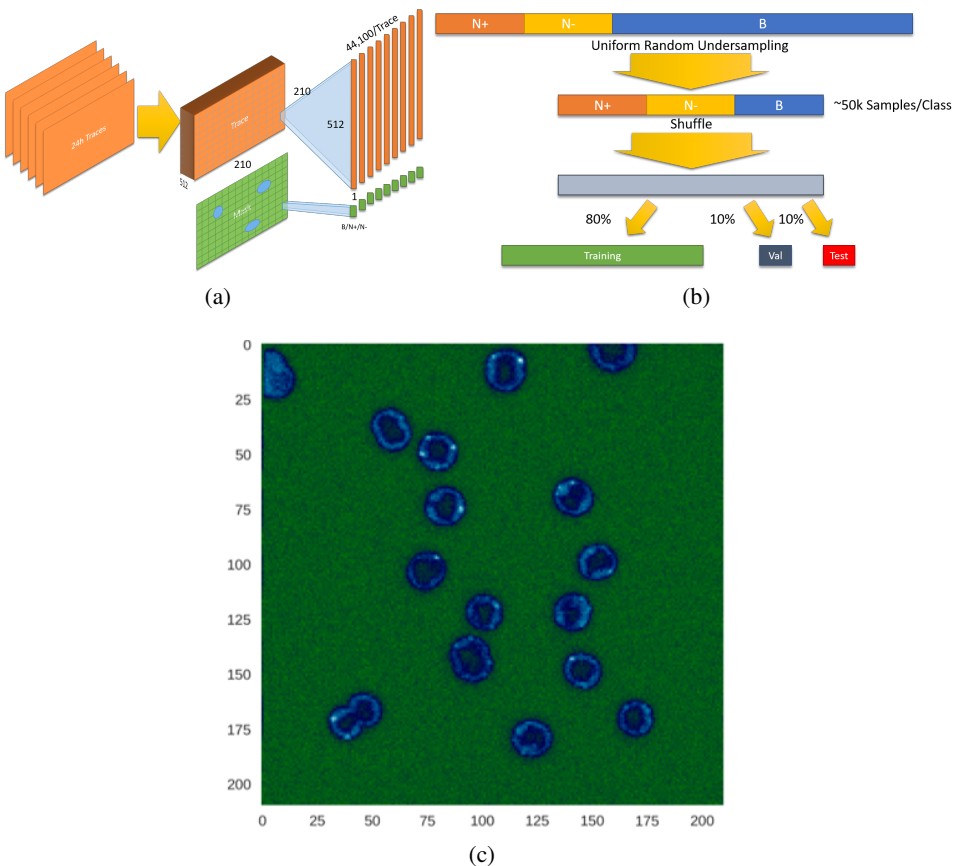

(a)                                              (b)

(c)

Figure 1: (a) Each original image and mask is split by pixel producing a dataset comprising 512-dimensional image vectors with an associated mask scalar. (b) Uniform random undersampling is used to balance the dataset before being undergoing a 80/10/10 Training/Validation/Test split. (c) Median values of example trace (5) from the 48 hour collection dataset.

## 2.4 ITERATIVE, DATA-DRIVEN SPARSE FEATURE SAMPLING

We use the densely connected neural network method described above to guide an iterative and data-driven sparse feature sampling algorithm. This approach has some similarities that in Li et al. (2011). However, we avoid computational overhead of an evolutionary algorithm by employing a greedy-type algorithm analyzing the synaptic weights of the neural network.

This method requires four discrete steps and a parameter $\tau$ which represents the decrease in accuracy which triggers a re-training.

1. Train an initial classifier neural network as in 2.3 on the set of all input features $\Omega_0$. The dense feed-forward neural network is similar to that in 2.3, however we adjust the network parameters to fit a shrinking input size and use an adadelta optimizer (Zeiler, 2012).

2. Compute the $\gamma(x_j) = \sum |w_i|$ where $w_i$ are the weights coming from $x_j$, a dimension in the input layer. The value $\gamma(x_j)$ acts as a metric for the 'worthiness' of an input feature.

3. Remove the dimension corresponding to the minimum $\gamma(x_j)$ to form $\Omega_{k+1}$ from $\Omega_k$. Determine the validation accuracy on $\Omega_{k+1}$ without retraining.

4. If the decrease in the accuracy is more than $\tau$, train a new network (possibly of smaller size to match $\Omega_{k+1}$) and repeat from Step 1. If not, repeat from Step 3. Alternatively, we can apply various halting conditions, e.g. a maximum number of iterations.

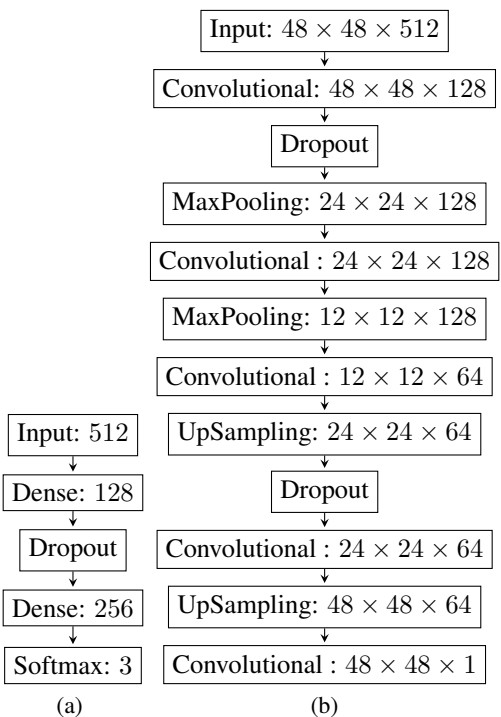

Figure 2: (a) The densely connected feed-forward neural network used in the classification task. (b) Convolutional neural network used for the masking task. All convolutional filters are $3 \times 3$ and pooling is done in $2 \times 2$ blocks.

## 2.5 CONVOLUTIONAL NEURAL NETWORK FOR CELL MASKING

Joint cell masking and classification was performed using neural networks to generate image masks highlighting the various cell types ($N^+$ or $N^-$). To accomplish this, we utilized a convolutional neural network similar to (Long et al., 2015) wherein the image undergoes a downsampling follwed by upsampling. The network architecture is shown in Fig. 2(b). All convolutional filters are $3 \times 3$, convolutional activation functions are rectified linear, and pooling/upsampling is done in $2 \times 2$ blocks.

Mean squared error acted as the loss function. The network was trained using an adadelta optimizer (Zeiler, 2012). As before, hyperparameter optimization was through the hyperas package.

## 2.6 COMPUTING HARDWARE

For training all neural networks, we used an Nvidia DGX-1 node. The DGX-1 is equipped with dual 20-core Intel Xeon ES-2698 CPUs, 512GB of system ram, and eight Nvidia Tesla 16GB P-100 GPUs with a total of over 28k CUDA cores.

## 3 RESULTS

### 3.1 CLASSIFICATION RESULTS

Our first task is to classify pixels as one of three categories Background (BG), Cell Grown in Nitrogen Containing Culture ($N^+$) or Cell Grown in Nitrogen Deplete Culture ($N^-$). We choose to do a per-pixel classification for several reasons. First, neural networks require a large number of training points, and by splitting our dataset into individual pixels we inflate the number of training samples. Second, a per-pixel classification can be accomplished using a simple densely connected multi-layer perception network. Hence, we can determine the type using the spectral without conflating spatial

information. The last reason is biological: In some applications, areas within a cell could be in different environments or states, and subcellular information may be desirable.

Densely connected feed-forward neural networks have a long history in pattern classification. Given the dimensionality of our data and the robustness of our training set, it is perhaps unsurprising that classification accuracy is high. Overall accuracy on the dataset is 98.9%, with details in Table 1.

Table 1: Precision, Recall, and F1-scores for the densely connected feed-forward network

|  | Precision | Recall | f1 |
|---|---|---|---|
| BG | 0.99 | 0.98 | 0.98 |
| $N^+$ | 0.99 | 1.00 | 0.99 |
| $N^-$ | 0.98 | 0.99 | 0.99 |
| Average | 0.99 | 0.99 | 0.99 |

This compares to roughly 97.8% accuracy using a random forest approach. As shown in Fig. 3(a), the majority of the error is due to mis-classifying BG-labeled pixels. This error is possibly due to the fact that the original ground truth masks were expert generated and thus some cells were excluded even though they have strong signal due to either being out of focus or being cut off by the edge of the image frame, see Fig. 3(c). Error between $N^+$ and $N^-$ pixels could be due to algorithm error or the effect of the $N^-$ condition is not uniform within or across cells. Indeed, future analysis may use similar methods to determine effect localization rather than classification.

One advantage of the feed-forward neural network approach is the ability to interpret the layer one weights, allowing our pruning method described in 2.4.

## 3.2 PRUNING DRAMATICALLY REMOVES UNNEEDED FEATURES

Although one of the benefits of hyperspectral imaging is the ability to sample many different wavelengths, there is both a time and resource cost associated with the spectral extent sampled. While there may be complex interactions across wavelengths, particularly in biological systems, it is expected that there would be considerable redundancy across input sensors. For cases in which the ultimate application of a hyperspectral system is classification as described above, we hypothesized that a reduced set of spectral frequencies could be identified that would be sufficient for application purposes. While this can be done *a priori* in some cases, our goal was to leverage an analytical method to identify the combination of reduced inputs necessary to achieve these results.

Accordingly, we next asked whether the neural network approach described in the above section could be used to down-select spectral features so as to enable classification with fewer input dimensions. As described in section 2.4, we used the trained pixel-level neural network representation to identify candidate dimensions—in this case wavelengths—that could be removed while preserving overall algorithmic accuracy.

As shown in Fig. 4(a), we were able to ignore many input dimensions from the images and still maintain highly effective classification. In effect, this shows that somewhere on the order of 90% of the frequencies sampled are not necessary for effective classification. This approach was incremental, while the least important dimensions could be safely ignored from the originally trained network, it was not surprising that the removal of more influential input channels (as identified by synpatic weights) required the networks to be retrained with the reduced inputs to maintain strong performance. However, the number of incremental training cycles was relatively low up until the network was highly reduced.

Fig. 4(b) shows when specific frequencies are removed through this pruning procedure (darker colors are those removed earlier). Not surprisingly, there is structure associated with which frequencies are removed first and which must be maintained for strong classification performance. Due to parameter sensitivity and variability during the training process, the features which are pruned and the order in which they are pruned are not unique.

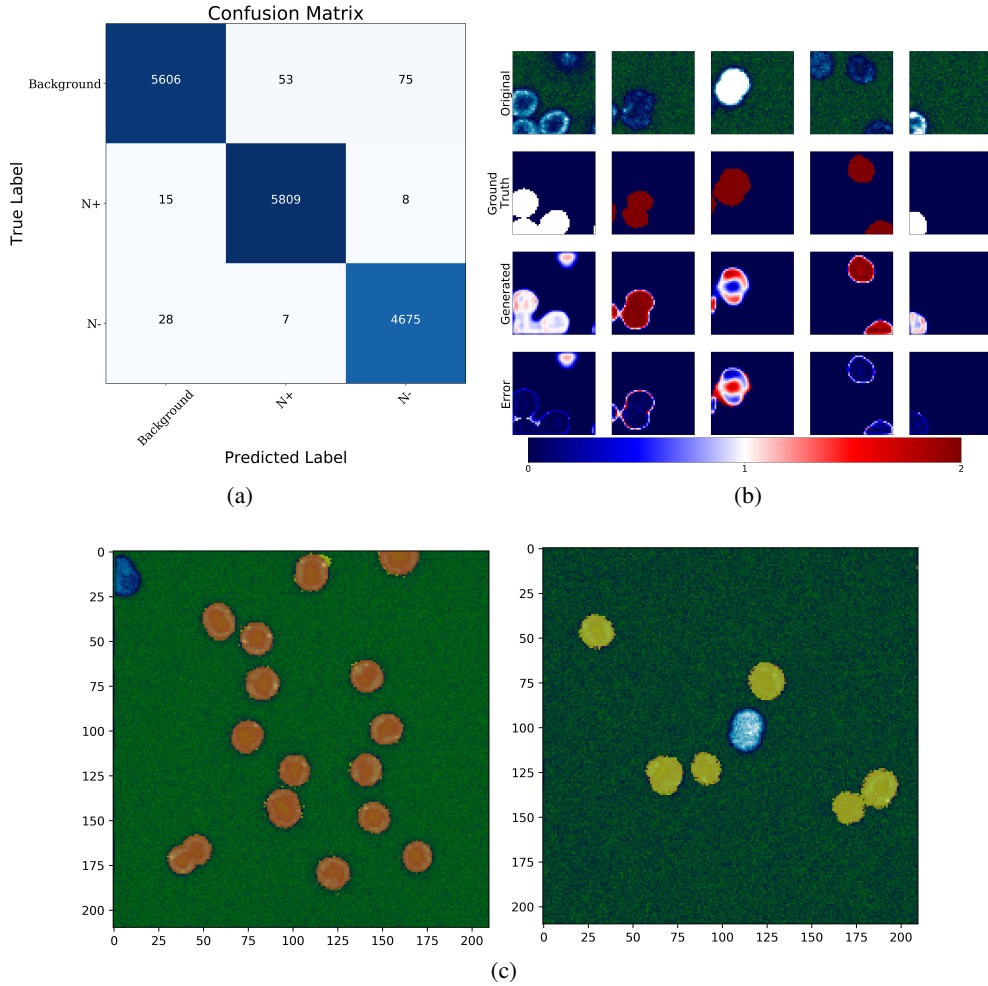

Figure 3: (a) Confusion matrix for the densely connected feed-forward neural network. (b) Sample crops of original 48hr images, ground truth masks, generated masks and error. BG is coded 0; $N^+$ is coded 1; $N^-$ is coded 2. (c) Sample images from 48hr with dense network pixel classification overlaid. $N^+$ is orange; $N^-$ is yellow; BG is transparent.

### 3.3 CNN EFFECTIVELY GENERATES MASKS

Finally, we examined whether our analysis approach could be extended to perform not only the experimental classification task but also the identification of regions of interest, namely cells, in our data. Because the pixel-based method described above eliminated the spatial structure of the data, we asked whether deep convolutional networks would be capable of jointly performing both classification and cell masking. Convolutional networks are state of the art in standard image classification and image segmentation tasks, so we expected them to be effective at the task of spatially identifying cells.

As shown in Fig. 3(b), the convolutional networks were able to generate a mask image simultaneously segmenting and classifying the cells. Over a 100-image test set generated from the 48hr dataset, the average per-pixel L1-error was 0.041. By allowing the network to produce non-integer values, we obtain smooth and detailed cell outlines. Furthermore, by conjoining the spatial and spectral dimensions, we expect this approach to be more robust to noise and extraneous objects.

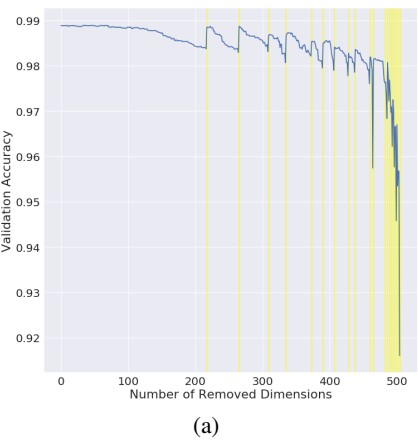
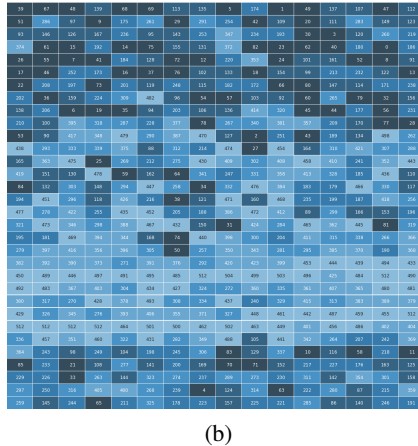

(a)                                         (b)

Figure 4: (a) The validation accuracy is plotted against the number of removed dimensions. Yellow vertical lines represent a point where the network is re-trained. The algorithm was halted after 20 re-training iterations; we set $\tau = 0.005$. (b) Individual frequencies labeled according to their removal order. Frequencies increase left-to-right, top-to-bottom. Darker colors are those removed earlier. Values of 512 represent frequencies remaining after the algorithm halted.

## 4  CONCLUSION

In this study, we demonstrate that modern deep artificial neural network approaches can be used to perform rapid classification of biological data sampled by hyperspectral imaging. Both the pixel-based and whole image-based classification results demonstrate that these approaches are highly effective with the class of data represented by this experimental data and suggest that deep neural network approaches are well suited for hyperspectral imaging analysis even in non-trivial application domains such as biological tissue.

We believe that the sampling reduction technique we describe here is a unique use of a neural network's classification ability to guide the identification of which particular sensors—in this case wavelengths—are necessary to measure. Most dimensionality reduction methods, such as PCA and non-linear variants such as local linear embedding (LLE), are focused primarily on reducing the size. While they can identify channels that are not used at all, they are more directed towards storing and communicating data in fewer dimensions which still leverage information sampled across the original cadre of sensors. Thus these dimensionality reduction do not necessarily reduce the demands on the sensor side, even though they do often compress and describe data quite effectively.

The methods described here share some similarities to existing techniques for hyperspectral imaging using techniques such as deep stacked autoencoders or principal components analysis coupled with a deep convolutional network that extract high-level features which can then be fed into a simple classifier (Chen et al., 2014; Zhao & Du, 2016). In contrast, our approach is focused on directly going from the data to the classification of either pixels or whole regions (in our case, cells). This allows us to better leverage the structure of the dimensionality of the data, which for hyperspectral scenarios is often sparser in absolute numbers of images but is proportionally richer in terms of dimensionality.

Given deep neural networks' history of broad applicability in other domains, we fully expect that these methods will be generalizable to other, similar datasets and anticipate subsequent analysis of a variety of cell types under experimental conditions. Further refinement of our convolutional neural network should provide effective and efficient sub-cellular segmentation via embedded computing platforms, and ultimately we aim to extend the use of these neural network algorithms to inform experimental results.

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
