# OpenReview forum: "Data-driven Feature Sampling for Deep Hyperspectral Classification and Segmentation"
_ICLR.cc/2018/Conference — Reject_

### Official Review · AnonReviewer2 · 2017-11-23
**Sequential backward feature selection using a mixture of filtering and wrapping, applied to multispectral data**

**Rating:** 3
**Confidence:** 5

**Review:**

Authors propose a greedy scheme to select a subset of (highly correlated) spectral features in a classification task. The selection criterion used is the average magnitude with which this feature contributes to the activation of a next-layer perceptron. Once validation accuracy drops too much, the pruned network is retrained, etc.

Pro:
- Method works well on a single data set and solves the problem
- Paper is clearly written
- Good use of standard tricks

Con:
- Little novelty

This paper could be a good fit for an applied conference such as the International Symposium on Biomedical Imaging.

---

### Official Review · AnonReviewer1 · 2017-11-26
**Review of "Data-driven Feature Sampling for Deep Hyperspectral Classification and Segmentation"**

**Rating:** 6
**Confidence:** 5

**Review:**

This paper explores the use of neural networks for classification and segmentation of hypersepctral imaging (HSI) of cells. The basic set-up of the method and results seem correct and will be useful to the specific application described. While the narrow scope might limit this work's significance, my main issue with the paper is that while the authors describe prior art in terms of HSI for biological preps, there is a very rich literature addressing HSI images in other domains: in particular for remote sensing. I think that this work can (1) be a lot clearer as to the novelty and (2) have a much bigger impact if this literature is addressed. In particular, there are a number of methods of supervised and unsupervised feature extraction used for classification purposes (e.g. endmember extraction or dictionary learning). It would be great to know how the features extracted from the neural network compare to these methods, as well as how the classification performance compares to typical methods, such as performing SVM classification in the feature space. The comparison with the random forests is nice, but that is not a standard method. Putting the presented work in context for these other methods would help make there results more general, and hopefully increase the applicability to more general HSI data (thus increasing significance).

An additional place where this comparison to the larger HSI literature would be useful is in the section where the authors describe the use of the network weights to isolate sub-bands that are more informative than others. Given the high correlation in many spectra, typically something like random sampling might be sufficient (as in compressive sensing). This type of compression which can be applied at the sensor -- a benefit the authors mention of their band-wise sub-sampling. It would be good to acknowledge this prior work and to understand if the features from the network are superior to the random sampling scheme.

For these comments, I suggest the authors look at the following papers (and especially the references therein):
[1] Li, Chengbo, et al. "A compressive sensing and unmixing scheme for hyperspectral data processing." IEEE Transactions on Image Processing 21.3 (2012): 1200-1210.
[2] Bioucas-Dias, José M., et al. "Hyperspectral unmixing overview: Geometrical, statistical, and sparse regression-based approaches." IEEE journal of selected topics in applied earth observations and remote sensing 5.2 (2012): 354-379.
[3] Charles, Adam S., Bruno A. Olshausen, and Christopher J. Rozell. "Learning sparse codes for hyperspectral imagery." IEEE Journal of Selected Topics in Signal Processing 5.5 (2011): 963-978.

---

### Official Review · AnonReviewer3 · 2017-12-03
**Incremental Technical Contribution**

**Rating:** 4
**Confidence:** 5

**Review:**

In this paper, the authors proposed a framework to classify cells and implement cell segmentation based on the deep learning
techniques. Using the classification results to guide the feature selection method, the method can achieve comparable performance even 90% of the input features are reduced.

In general, the paper addresses an interesting problem, but the technical contribution is rather incremental. The authors seem to apply some well-define methods to realize a new task. The authors are expected to clarify their technical contributions or model improvement to address the specific problem.

Moreover, there also exist some recent progress on image segmentation, such as FCN or mask R-CNN. The authors are expected to demonstrate the results by improving these advanced models.

In general, this is an interesting paper, but would be more fit to MICCAI or ISBI.

---

### Decision · Program_Chairs · 2018-01-29
**ICLR 2018 Conference Acceptance Decision**

**Decision:**

Reject

**Comment:**

Area chair is in agreement with reviewers: this is a good experiment that successfully applies specific machine learning techniques to the particular task. However, the authors have not discussed or studied the breadth of other possible methods that could also solve the given task ... besides those mentioned by the reviewers, U-Nets, and variants thereof, come to mind. Without these comparisons, the novelty and significance cannot be assessed.

Authors are encouraged to study similar works, and perform a comparison among multiple possible approaches, before submission to another venue.